# SARS-CoV-2 mRNA Vaccines Induce Cross-Reactive Antibodies to NL63 Coronavirus but Do Not Boost Pre-Existing Immunity Anti-NL63 Antibody Responses

**DOI:** 10.3390/vaccines13030268

**Published:** 2025-03-04

**Authors:** Weiyi Tang, Zi Wei Chang, Yun Shan Goh, Yong Jie Tan, Pei Xiang Hor, Chiew Yee Loh, David C. Lye, Barnaby E. Young, Lisa F. P. Ng, Matthew Zirui Tay, Laurent Rénia

**Affiliations:** 1A*STAR Infectious Diseases Labs (A*STAR ID Labs), Agency for Science, Technology and Research (A*STAR), Singapore 138648, Singapore; tang_weiyi_from.tp@idlabs.a-star.edu.sg (W.T.); chang_zi_wei@idlabs.a-star.edu.sg (Z.W.C.); goh_yun_shan@IDLabs.a-star.edu.sg (Y.S.G.); tan_yong_jie@idlabs.a-star.edu.sg (Y.J.T.); hor_pei_xiang@idlabs.a-star.edu.sg (P.X.H.); loh_chiew_yee@idlabs.a-star.edu.sg (C.Y.L.); lisa_ng@idlabs.a-star.edu.sg (L.F.P.N.); matthew_tay@idlabs.a-star.edu.sg (M.Z.T.); 2National Centre for Infectious Diseases, Singapore 308442, Singapore; david_lye@ncid.sg (D.C.L.); barnaby_young@ncid.sg (B.E.Y.); 3Lee Kong Chian School of Medicine, Nanyang Technological University, Singapore 308232, Singapore; 4School of Biological Sciences, Nanyang Technological University, Singapore 637551, Singapore

**Keywords:** SARS-CoV-2, mRNA vaccine, NL63, BNT162b, mRNA-1273

## Abstract

Background/Objectives: mRNA vaccines have demonstrated strong immunogenicity and efficacy against SARS-CoV-2. However, the extent of antibody cross-reactivity against human seasonal coronaviruses, such as NL63, remains unclear. Furthermore, it is unknown whether pre-existing antibody responses against NL63 might influence the outcome of SARS-CoV-2 mRNA vaccination. Methods: We used a flow cytometry-based serological assay and an in vitro neutralization assay to analyze NL63 antibody responses in sera from SARS-CoV-2 mRNA-vaccinated mice and plasma samples from a vaccinated human cohort. Results: We found that the Moderna mRNA-1273 vaccine can generate cross-reactive antibodies against NL63. Importantly, SARS-CoV-2 mRNA vaccination did not boost pre-existing anti-NL63 responses in humans, and pre-existing NL63 antibody levels did not affect the antibody response induced by SARS-CoV-2 mRNA vaccination. Conclusions: These findings suggest that while SARS-CoV-2 mRNA vaccination can induce cross-reactive antibodies against NL63, pre-existing immunity to this seasonal coronavirus does not appear to significantly impact vaccine immunogenicity. These findings contribute to our understanding of the complex interplay between pre-existing immunity to seasonal coronaviruses and the immune response generated by SARS-CoV-2 mRNA vaccines.

## 1. Introduction

The COVID-19 pandemic has highlighted the urgent need for effective vaccines to fight coronavirus infections. Messenger RNA (mRNA) vaccines have demonstrated strong immunogenicity and high efficacy against COVID-19 severe diseases. However, the effect of SARS-CoV-2 mRNA vaccines on other coronaviruses, including common cold-causing human seasonal coronaviruses like NL63, remains largely unknown [1,2]. NL63 (HCoV-NL63) causes respiratory infections in humans, which can be severe in vulnerable populations such as children or immunocompromised individuals [1]. Although NL63 belongs to a different coronavirus genus (alphacoronavirus) than SARS-CoV-2 (betacoronavirus), both viruses use the ACE2 receptor for host cell entry, raising the question of whether SARS-CoV-2 vaccination might induce cross-reactive immunity against NL63 [3]. Understanding this potential cross-reactivity is important for a comprehensive evaluation of the immunological impact of SARS-CoV-2 mRNA vaccines. Although initial data regarding the impact of SARS-CoV-2 mRNA vaccines on other coronaviruses, such as NL63, were limited, recent studies have begun to examine their effects, particularly in relation to antibody and T cell cross-reactivity [4,5,6,7].

Here, we have investigated whether SARS-CoV-2 mRNA vaccination generates cross-reactive antibodies against NL63 and how this may affect pre-existing anti-NL63 antibody responses. We have also examined whether pre-existing NL63 antibody levels influence the antibody response induced by SARS-CoV-2 mRNA vaccination. Our findings provide valuable insights into the complex interplay between SARS-CoV-2 vaccination and immunity to seasonal coronaviruses, with implications for future vaccine strategies and public health policies.

## 2. Materials and Methods

### 2.1. Mice

Six- to eight-week-old female C57BL/6 mice, purchased from Invivos (Singapore), were used in this study. The mice were housed in a specific pathogen-free environment at the A*STAR Biological Resource Centre (BRC), Singapore. All experiments and procedures were performed with approval from the Institutional Animal Care and Use Committee (IACUC), IACUC # 211673, in accordance with the Animal & Veterinary Service (AVS) and the National Advisory Committee for Laboratory Animal Research (NACLAR) of Singapore. BNT162b2 (Pfizer, New York, NY, USA) and mRNA-1273 (Moderna, Cambridge, MA, USA) vaccines (approved for use by the Ministry of Health, Singapore), were obtained from the Ministry of Health for this study. Mice were immunized intramuscularly with 1 µg of either BNT162b2 or mRNA-1273 diluted in phosphate-buffered saline (PBS) to a final volume of 50 μL. This two-dose immunization regimen (day 0 and day 21) followed established protocols from previous studies [8,9]. Control mice received PBS only (Figure 1a). Blood samples were collected via retro-orbital sampling on day 28 to assess humoral responses. For NL63 virus immunization, 7 × 10⁴ TCID50/50 μL of purified NL63 in PBS was mixed with 50 μL of complete Freund’s Adjuvant (first dose) or incomplete Freund’s Adjuvant (subsequent doses) and injected subcutaneously into each mouse on days 0, 21, 35, and 49. Blood samples were collected via retro-orbital sampling, from anesthetized mice, on day 56 after the first dose to assess humoral responses.

### 2.2. Ethics Statement and Study Population

Written and signed informed consent was obtained from participants in accordance with the tenets of the Declaration of Helsinki. Serum and plasma samples were collected in two different studies previously reported. Fifteen samples came from the PROTECT study where individuals received 3 doses of BNT162b2 vaccines. The second dose was injected 3 weeks after the first dose, and the third dose was injected one year after. Samples were collected on day 0 (before vaccination), 3 months, 6 months after second dose and 1 to 3 months after third dose (Figure 2a) [10]. The PROTECT study design and protocol were previously reviewed and approved by the National Healthcare Group Institutional Review Board, Domain-specific review board, Singapore (DSRB #2012/00917, approval granted the 1 September 2012 and renewed the 28 December 2023). Fifteen samples came from the PRIBIVAC study where participants were previously recruited and enrolled for a clinical trial at the National Centre for Infectious Diseases in Singapore for a clinical trial studying prime-boost vaccination strategies (ClinicalTrials.gov identifier: NCT05142319). The trial design and first results of this trial have been described previously [11,12]. This study was reviewed and approved by the National Healthcare Group Institutional Review Board, Domain-specific review board, Singapore (DSRB #2021/00821, approval granted the 21 October 2021). In the PRIBIVAC study, individuals were selected if they had received two doses of the BNT162b2 vaccine between 174 and 327 days (median 228 days) prior to enrolment. Participants were excluded if they had: previous SARS-CoV-1 or SARS-CoV-2 infection (confirmed by the absence of antibodies to the SARS-CoV-2 nucleocapsid protein), a history of being immunocompromised (e.g., requiring immunosuppressant medication, undergoing chemotherapy, or diagnosed with leukemia). Each participant was given one dose of mRNA-1273 booster after enrollment. Samples were collected before mRNA-1273 booster, and 7 days and 1 month after booster (Figure 2a).

### 2.3. Virus

The NL63 virus was obtained from BEI Resources (#NR-470) (ATCC, Washington, DC, USA). NL63 was cultured in the LLC-MK2 cell line under 33 °C in DMEM/F-12 medium (Thermo Fischer Scientific, Singapore). For virus purification, virus supernatant was collected when ~50% CPE was observed in the culture. A volume of 30 mL of supernatant was added to each Ultra-Clear Centrifuge Tube (Beckman Coulter Life Sciences, Singapore). A volume of 4 mL of 20% sucrose in 1 × TNE was slowly added to the bottom of each tube. The tubes were then placed in ultracentrifuge cases and loaded to a Model L-100 k Series Ultracentrifuge (Beckman Coulter). The virus was ultracentrifuged at 28,000 rpm for 4 h at 4 °C, with maximum acceleration and no-break deceleration. After ultracentrifugation, the bulk supernatant in each tube was removed by decanting, and tubes were kept inverted on a paper towel for drip drying. A volume of 100 μL of PBS was added to resuspend and wash the virus, and the tube was kept vertically on ice for 30 min. The viral preparations in all tubes were mixed and kept under −80 °C before usage.

### 2.4. Spike Protein Flow Cytometry-Based (SFB) Assay

The SFB assay was performed as previously described [13]. HEK293T cells, engineered to stably express the NL63 spike protein on their surface, were seeded at a density of 2.5 × 10⁵ cells per well in 96-well V-bottom plates (Thermo Scientific). Cells were incubated with human plasma or mouse serum samples (diluted 1:500 in PBS containing 10% fetal bovine serum) for 30 min at 4 °C. This was followed by secondary staining with Alexa Fluor 647-conjugated anti-human IgG (for human samples) or Alexa Fluor 647-conjugated anti-mouse IgG (for mouse samples) at a 1:700 dilution (Thermo Scientific), and propidium iodide (1 μg/mL; Sigma Aldrich, Singapore) for 30 min at 4 °C. Cells were washed with PBS containing 10% fetal bovine serum after each staining step. Flow cytometry data were acquired using a CytoFLEX Flow Cytometer (Beckman Coulter) and analyzed with FlowJo software version 10.10. The cell population of interest was gated based on forward scatter area (FSC-A) versus side scatter area (SSC-A). The percentage of gated cells positive for Alexa Fluor 647 and proprium iodide negative (excluding dead cells) was used to quantify the level of antibodies binding to the NL63 or the SARS-CoV-2 spike proteins [14].

### 2.5. Neutralization Assay

Eighteen thousand LLC-MK2 cells were seeded into each well of a 96-well, black, clear-bottom plate (Corning, Charlotte, NC, USA) one day prior to the assay. NL63 virus (2000 TCID50/mL) was serially diluted in Dulbecco’s Modified Eagle Medium/Nutrient Mixture F-12 (DMEM/F12) containing 2% fetal bovine serum. Equal volumes of the diluted virus and serially diluted reagents were mixed and incubated at 33 °C for 1 h with three replicates per dilution (*n* = 3). The cell culture media was removed, and 100 μL of the virus-reagent mixture was added to each well. After a 7-day incubation, cell viability was measured using the CellTiter-Glo Luminescent Cell Viability Assay (Promega, Singapore) and a GloMax Explorer plate reader (Promega), according to the manufacturer’s instructions. NL63 typically requires at least 6 days to cause observable cytopathic effects (CPEs) in cell culture, and similar conditions have been described previously [3]. In our experiments, 6 days of incubation resulted in minimal CPE, with 7 days proving optimal. A “no virus” control was included in each experiment to assess cell viability using the CellTiter-Glo kit. The EC50, corresponding to the antibody dilution that inhibits 50% of viral infection, was then calculated.

### 2.6. Statistical Analysis

Statistical analyses were performed using GraphPad Prism 7 software. Kruskal–Wallis tests, followed by Dunn’s multiple comparison post hoc tests, were used for comparisons between multiple groups. Friedman tests, with Dunn’s multiple comparison post hoc tests, were used for comparisons between multiple time points. A *p*-value of less than 0.05 was considered statistically significant. Spearman’s correlation analysis was used to assess the correlation between antibody responses against the wild-type SARS-CoV-2 spike protein and the NL63 spike protein. Least squares regression was used to analyze the neutralization assay results.

## 3. Results

### 3.1. SARS-CoV-2 mRNA Vaccine Can Generate Cross-Reactive Antibody Against Human Seasonal Coronavirus NL63

We first conducted experiments in mice, as it is difficult to confirm the absence of prior exposure to seasonal coronaviruses like NL63 in adult humans. Mice were immunized with two doses of either the BNT162b2 or mRNA-1273 vaccine, with a 3-week interval between doses. Serum samples were collected 7 days after the second dose (Figure 1a). IgG antibody levels against the spike proteins of SARS-CoV-2 and NL63 were then measured using the SFB assay (Figure 1b) [13,14]. mRNA-1273 vaccination induced a significantly higher level of anti-NL63 spike antibodies (Figure 1c). This suggests that these vaccines might be capable of inducing cross-reactive and neutralizing antibodies against coronaviruses. While an increase in anti-NL63 spike antibodies was also observed after BNT162b2 vaccination, this increase was not statistically significant (*p* = 0.17). A live virus NL63 neutralization assay showed that the cross-reactive antibodies generated by mRNA-1273 vaccination had detectable albeit weak neutralizing activity against NL63 (Figure 1d).

Taken together, these results demonstrate that SARS-CoV-2 mRNA vaccination, particularly with mRNA-1273, can elicit a cross-reactive antibody response against NL63.

**Figure 1 vaccines-13-00268-f001:**
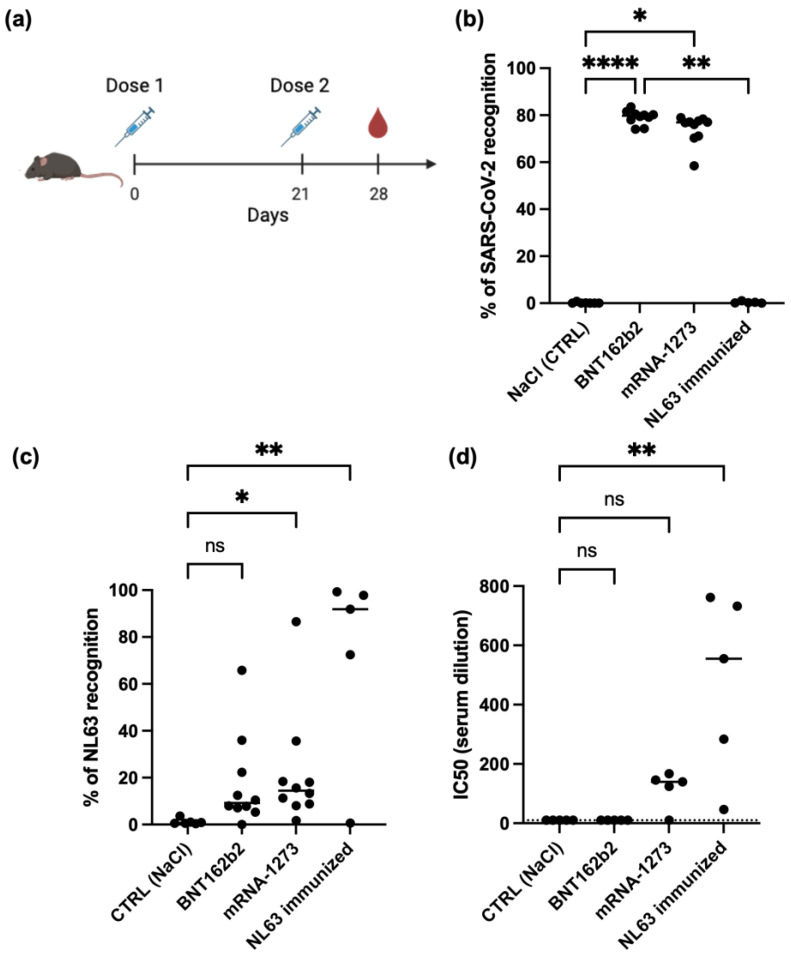
Cross-reactive anti-SARS-CoV-2 antibodies against NL063 induced by mRNA vaccines. (**a**) Mouse vaccination regime. (**b**,**c**) Antibody responses against SARS-CoV-2 WT (**b**) and NL63 (**c**) measured by SFB assay. Serum samples were collected after immunization with BNT162b2 (n = 10), mRNA-1273 (*n* = 10), heat-inactivated NL063 virus (*n* = 5), or NaCl as control (*n* = 5). Data are expressed as the percentage of SARS-CoV-2 or NL63 spike protein recognition, which corresponds to percentage of cells expressing SARS-CoV-2 or NL63 spike proteins that are recognized by antibodies specific for these proteins. (**d**) Neutralization antibody titers against NL063 in immunized mouse sera (*n* = 5). Median values are indicated by a line. ns: not significant, *: *p* < 0.05, **: *p* < 0.01, ****: *p* < 0.0001; by Kruskal–Wallis test.

### 3.2. SARS-CoV-2 mRNA Vaccine Cannot Boost Pre-Existing NL63 Antibody Responses

Samples from 15 individuals who received three doses of the BNT162b2 mRNA vaccine were assessed for IgG antibody responses against NL63 using the SFB assay (Figure 2a). For the PROTECT group, the median age was 33 years (95% CI of median 29–39; range 23–44), and volunteers were predominately female (93.3%) and Filipino (73.3%). For the PRIBIVAC group, the median age was 36 years (95% CI of median 31–59; range 23–76), and volunteers were predominantly Chinese (86.7%). Pre-existing NL63 antibodies were prevalent in human plasma samples before vaccination, with a median spike recognition of 17.5% (Figure 2b). However, after two doses of BNT162b2, there was no significant change in anti-NL63 antibody levels. Furthermore, anti-NL63 levels decreased after the third BNT162b2 dose. This observation is consistent with the decline in neutralizing antibody titers against NL63 after three doses of BNT162b2 (Figure 2c). This suggests a waning of the anti-NL63 antibody response in absence of boosting of NL-63 exposure and further confirms that the BNT162b2 vaccine has no significant influence on anti-NL63 antibody responses.

In the next experiment, samples from 10 more individuals who had previously received two doses of the BNT162b2 vaccine and had received a booster dose of the mRNA-1273 vaccine (Figure 2a) were tested. Neither the antibody response against NL63 nor the NL63 neutralizing antibody titer changed after the mRNA-1273 booster dose (Figure 2d,e).

**Figure 2 vaccines-13-00268-f002:**
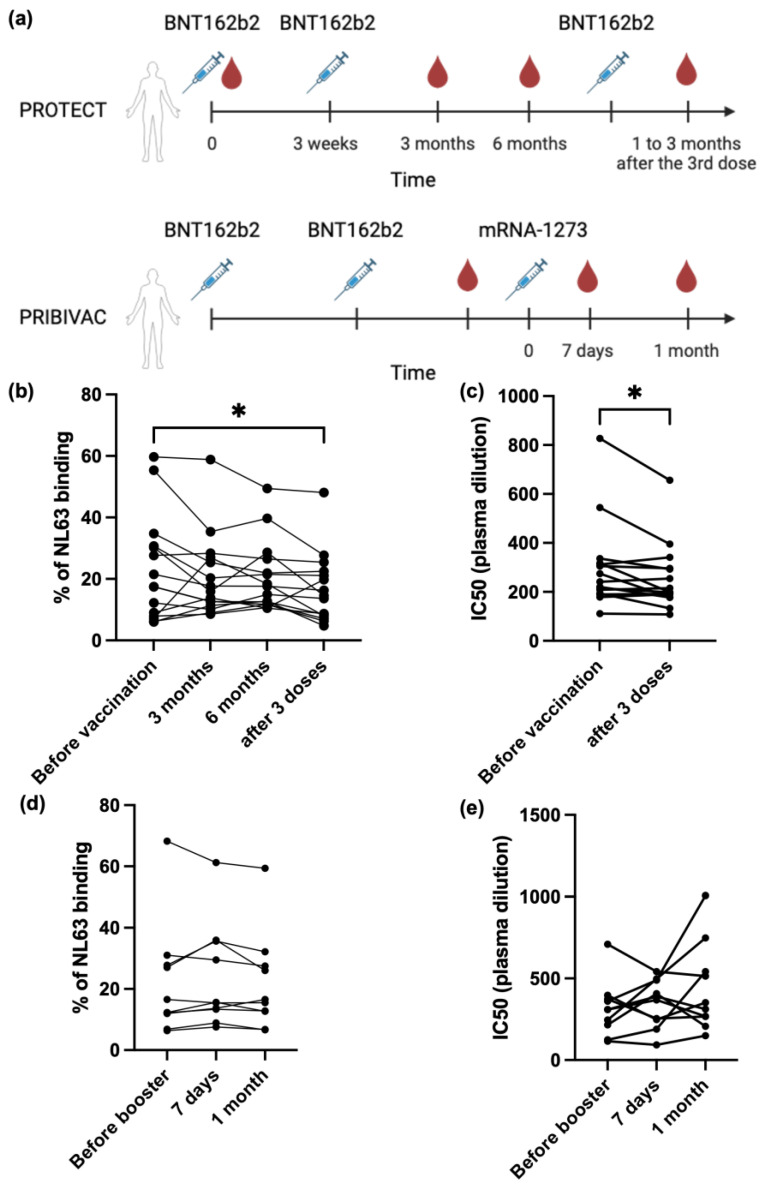
No boosting effect of NL63 antibody after SARS-CoV-2 mRNA vaccination. (**a**) Vaccination regime for individuals recruited in this study. (**b**,**d**) Antibody response against NL63 for PROTECT (**b**) (*n* = 15) and PRIBIVAC (**d**) (*n* = 10) groups measured by SFB assay. Data are expressed as percentage of NL63 spike protein recognition, which corresponds to percentage of cells expressing NL63 spike proteins that are recognized by antibodies specific for these proteins. (**c**,**e**) Neutralization antibody titer against NL63 in PROTECT (**c**) (n = 15) and PRIBIVAC (**e**) (*n* = 10) groups. *: *p* < 0.05; by Kruskal–Wallis test (**b**) or Wilcoxon test (**c**).

### 3.3. NL63 Pre-Existing Immune Response Did Not Affect Anti SARS-CoV-2 Antibody Response Imduced by Vaccination

To determine whether pre-existing antibody responses against NL63 affected SARS-CoV-2 vaccine immunogenicity, we analyzed the correlation between pre-existing anti-NL63 IgG levels and SARS-CoV-2 IgG antibody levels at different time points after vaccination with mRNA BNT162b2 vaccine alone or the prime-boost combination of the BNT162b2 vaccine followed by mRNA-1273. We found no significant correlation between NL63, and SARS-CoV-2 IgG levels (Figure 3), indicating that pre-existing NL63 antibody levels did not influence the priming, boosting, longevity, or waning of the antibody response in individuals receiving the BNT162b2 vaccine alone or the prime-boost combination of the BNT162b2 vaccine followed by mRNA-1273.

## 4. Discussion

Previous studies have shown that infections with the seasonal coronavirus NL63 are common in the human population [1]. These infections have been hypothesized to influence immune responses to subsequent SARS-CoV-2 infection or vaccination due to pre-existing antibodies binding to vaccine antigen and influencing processes such as antigen clearance and antigen presentation. Reciprocally, SARS-CoV-2 vaccination may also influence immune responses against seasonal coronaviruses (Table 1). Most current research has focused on cross-reactivity among individuals recovering from COVID-19 rather than on vaccination outcomes. According to the published literature, SARS-CoV-2 infection generally does not generate cross-reactive antibodies against NL63 (Table 1). While some studies have reported higher anti-NL63 antibody levels in COVID-19 patients compared to healthy individuals, this effect is likely due to the boosting of pre-existing anti-NL63 memory responses rather than the generation of cross-reactive anti-SARS-CoV-2 antibodies [15]. This is supported by the observation that there is no correlation between anti-SARS-CoV-2 and anti-NL63 antibody levels after infection [16].

To address these questions, we used two experimental approaches. First, we immunized mice with SARS-CoV-2 mRNA vaccines (both Pfizer and Moderna). This approach was chosen because most humans have been exposed to seasonal coronaviruses and may possess residual immune memory, even if they are seronegative. A recent study has shown that an adenovirus vaccine (FAdV-9-S19) generated low-level protection against NL63 and OC43 in K18-hACE2 mice [39]. Here, we found that immunization with both mRNA vaccines generated cross-reactive antibodies against the human seasonal coronavirus NL63. However, only the mRNA-1273 (Moderna) vaccine induced neutralizing antibodies. This suggests that the two mRNA vaccines elicit different antibody repertoires, with the mRNA-1273 vaccine inducing more cross-neutralizing antibodies against NL63.

Although these two vaccines share the same amino acid sequence for the spike protein [7], they differ in their lipid nanoparticle formulations and mRNA modifications. These differences may affect immunogenicity and the breadth of the antibody response, as has been seen previously when assessing breadth against variants of SARS-CoV-2 [7].

The observed cross-reactivity between NL63 and SARS-CoV-2 spike proteins (included in some vaccines), despite their classification in different genera (alpha- and beta-coronavirus, respectively), raises intriguing questions. Cross-reactivity is typically more readily observed within the same genus. The spike protein’s receptor-binding domain (RBD) is required for viral entry. Although sequence similarity between NL63 and SARS-CoV-2 RBDs is low (~25%) [40], certain critical residues involved in receptor binding may be conserved or exhibit similar structural positioning [41]. Even limited similarity in key functional regions can, in some cases, elicit cross-reactive antibodies [40]. However, the overall sequence divergence suggests that RBD similarity alone is unlikely to explain the observed cross-reactivity fully. Antibodies targeting epitopes in other parts of the spike protein, such as the more conserved S2 subunit, may also contribute. These regions might exhibit greater sequence conservation between NL63 and SARS-CoV-2, leading to cross-reactive antibody responses. Conserved sequences containing shared epitopes targeted by neutralizing antibodies have been reported. Still, their importance in anti-coronavirus immunity requires further validation, particularly given the implications for pan-coronavirus vaccine development [16,40]. Notably, broadly neutralizing monoclonal antibodies recognizing the spike protein fusion peptide of diverse coronaviruses, including SARS-CoV-2 and NL63, have been reported [36].

In the second set of experiments, we investigated the effect of SARS-CoV-2 mRNA vaccination on pre-existing anti-NL63 responses in humans. We observed that vaccination did not boost NL63 antibody responses, and that these anti-NL63 responses did not affect SARS-CoV-2 antibody level induced by vaccination. Previous studies have also reported that the immune imprinting of other coronaviruses was not boosted after vaccination of the inactivated COVID-19 vaccine [42], and that pre-existing immunity against seasonal coronaviruses had no negative effect for SARS-CoV-2 infection in mice [4]. One limitation of this study is that we did not have access to a cohort of mRNA-1273-only vaccinated humans. In our cohort, individuals received two BNT162b2 doses and one mRNA-1273 booster dose. However, it was reported elsewhere that two doses of mRNA-1273 vaccination can boost NL63 antibody response compared to one dose of vaccination [43]. Therefore, we postulate that one dose of mRNA-1273 booster vaccination dose is not sufficient to boost cross-reactive anti-NL63 response.

An additional limitation of this study is that only cross-reactive antibody response was evaluated. It was reported elsewhere that when T cell responses were assessed, SARS-CoV-2 vaccination or exposure boosted cross-reactive T cell responses against other coronaviruses (Table 1). One study showed that SARS-CoV-2 mRNA vaccination increased the T cell response to NL63 three-fold, while no responses to other seasonal coronaviruses were boosted [5]. Another study found that cross-reactive T cell responses were higher in seronegative healthcare workers than healthy individuals, but the level decreased after SARS-CoV-2 infection, suggesting that exposure to SARS-CoV-2 might interfere with the cross-reactive T cell responses [44]. On the other hand, pre-existing T cell responses to SARS-CoV-2 have also been studied. These cross-reactive T cells widely existed before SARS-CoV-2 infection, and their epitopes can target various SARS-CoV-2 proteins, such as the spike, nucleocapsid, and RTC [45,46,47]. Interestingly, one study identified the highly conserved spike fusion peptide sequence as a T cell epitope, suggesting fusion peptide as a potential vaccine candidate to generate broad immune responses among coronaviruses for both B cell and T cell activation [45]. Another study identified S_1200–1226_ as the epitope for pre-existing cross-reactive T cells from healthy donors, which spanned across the heptad repeat 2 (HR2) and the transmembrane (TM) segment [48]. Additional studies in a small number of individuals suggested that pre-existing CD4+ and CD8+ T cells against alpha common cold coronavirus (229E) but no other seasonal coronaviruses were associated with protection from symptomatic and fatal SARS-CoV-2 infections in unvaccinated COVID-19 patients [49].

## 5. Conclusions

mRNA vaccines are effective tools against SARS-CoV-2 infections, but their ability for a broad protection have not been fully studied. The effect of BNT162b2 and mRNA-1273 vaccinations on the cross-reactive anti-NL63 were evaluated. In addition, the pre-existing anti-NL63 responses did not affect SARS-CoV-2 antibody level induced by SARS-CoV-2 vaccination.

## Figures and Tables

**Figure 3 vaccines-13-00268-f003:**
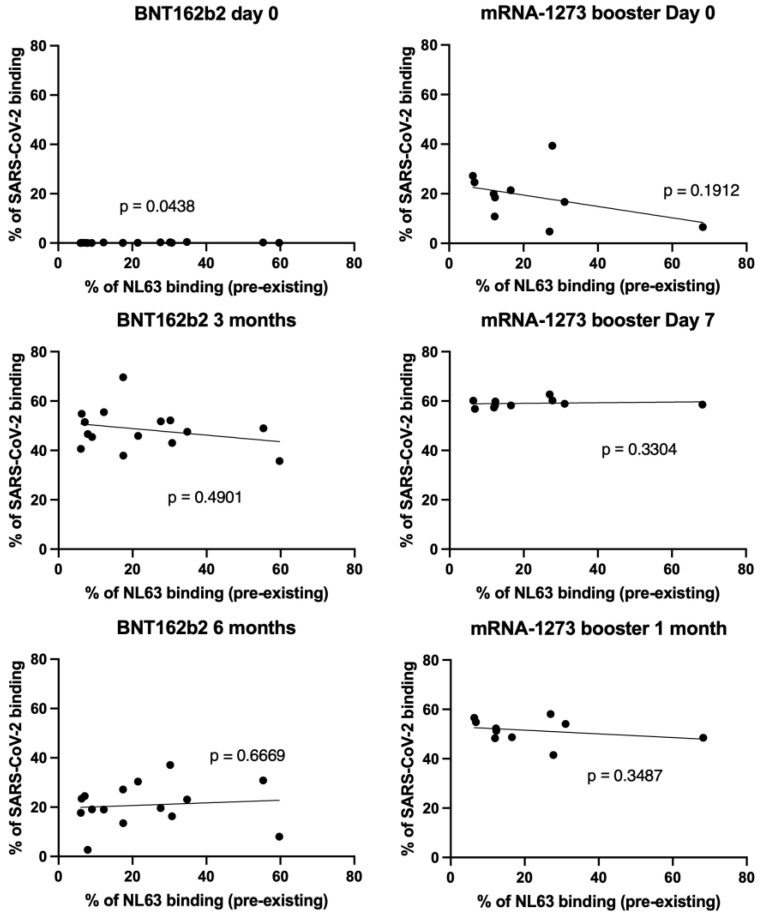
Correlation between pre-existing NL63 antibody levels and SARS-CoV-2 antibody levels after BNT162b2 vaccination (*n* = 15) or mRNA-1273 booster (*n* = 10). No response to SARS-CoV-2 was observed on day 0 for BNT162b2, and hence it is not included in the analysis. *p*-value was calculated by Spearman correlation.

**Table 1 vaccines-13-00268-t001:** Previous studies on whether SARS-CoV-2 infection or vaccination affects NL63 antibody response.

Experiment Group	Control Group	Antibody or T Cells?	Tested Antigen on NL63	Result	Ref.
PCR-positive	PCR-negative close contacts	IgG antibody	RBD	No difference	[8]
PCR-positive	Pre-COVID-19 samples	IgG antibody	RBD	No difference	[9]
COVID-19 severe	COVID-19 mild	IgG antibody	S and N	No difference	[17]
COVID-19 convalescents	Healthy individuals	IgG antibody	S1	No difference	[18]
COVID-19 convalescents	Healthy individuals	IgG antibody	S and N	No difference	[19]
COVID-19 convalescents	Healthy individuals	IgG antibody	N	No difference	[20]
COVID-19 convalescents	Healthy individuals	IgG antibody	S	No difference	[21]
COVID-19 convalescents	Healthy individuals	IgG antibody	RBD	No difference	[22]
COVID-19 patients	Healthy individuals	IgG antibody	Pseudovirus	Higher neutralization	[23]
COVID-19 convalescents	Pre-COVID-19 samples	IgG antibody	S2 and N	Higher response	[24]
COVID-19 convalescents	Healthy individuals	IgG antibody	N	Higher response	[25]
COVID-19 convalescents	COVID-19 admission	IgG antibody	S and N	Higher against N in severe patients	[8]
COVID-19 convalescents	Healthy individuals	IgG antibody	S	Higher response	[16]
COVID-19 convalescents	Healthy individuals	IgG mAbs	S	3 strongly cross-reactive mAbs	[26]
Healthcare workers with direct/indirect contact to COVID-19 patients	Healthcare workers with no contact to COVID-19 patients	IgG, IgM and IgA antibody	S	Higher IgM between direct and no contact group	[27]
COVID-19 convalescents or BNT162b2 vaccination (2 doses)	Pre-COVID-19 samples	IgG antibody	S and N	No difference	[28]
AZD1222 primary vaccination and/or booster	Placebo	IgG antibody	S	No difference	[29]
After 1-dose vaccination (did not mention what vaccine was used)	Before vaccination (same individuals)	IgG antibody	Pseudovirus	Higher neutralization	[30]
BNT162b2 or mRNA-1273 vaccinated	Healthy individuals	IgG antibody	S	No difference	[15]
BNT162b2 vaccination (2 and 3 doses)	Before vaccination (same individuals)	IgG antibody	S1	No difference	[31]
BNT162b2 vaccination (2 doses)	Before vaccination (same individuals)	IgG antibody	S	No difference	[32]
AZD1222, BNT162b2 or mRNA-1273 vaccinated (2 doses)	AZD1222, BNT162b2 or mRNA-1273 vaccinated (1 dose)	IgG antibody	Pseudovirus	Higher neutralization for mRNA-1273	[33]
BNT162b2 or mRNA-1273	Healthy individuals	IgG antibody	S	No difference	[34]
BNT162b2 or mRNA-1273 (2 doses)	Before vaccination (same individuals)	IgG antibody	S	No difference	[35]
BBIBP-CorV vaccinated (2 doses)	Before vaccination (same individuals)	IgG antibody	S	Higher response	[6]
Infected, Ad26-vaccinated or DNA-vaccinated + re-infected macaques	Before treatment (same animals)	IgG antibody	S	Higher response	[36]
COVID-19 convalescents	Healthy individuals	CD4+ T cells	Pool peptide library S1 and S2	No difference	[37]
COVID-19 exposed individuals	Healthy individuals	CD4+ T cells	Pool peptide library S1, S2, M and NP	Higher against S1, M and NP	[38]
BNT162b2 or mRNA-1273 vaccinated	Healthy individuals	CD4+ T cells	Pool peptide library (whole virus)	Higher response	[5]
COVID-19 convalescents	Healthy individuals	CD8+ T cells	Homologous N_105–113_ (PPKVHFYYL)	No difference	[32]

## Data Availability

All the data and materials can be found in the manuscript. Other requirements can be obtained by contacting the author.

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
