# Peer review of "SARS-CoV-2 mRNA Vaccines Induce Cross-Reactive Antibodies to NL63 Coronavirus but Do Not Boost Pre-Existing Immunity Anti-NL63 Antibody Responses"

_vaccines, 2025, doi:10.3390/vaccines13030268_

Round 1
Reviewer 1 Report
Comments and Suggestions for Authors
The article investigates the potential cross-reactivity of SARS-CoV-2 mRNA vaccines (BNT162b2 and mRNA-1273) against the human seasonal coronavirus NL63. The study explores the following points:
- Cross-Reactive Antibodies: The research found that SARS-CoV-2 mRNA vaccination (particularly with mRNA-1273) can generate antibodies that cross-react with NL63, suggesting that these vaccines might trigger immune responses against multiple coronaviruses. The mRNA-1273 vaccine was more effective at producing cross-reactive antibodies, which also exhibited some neutralizing activity against NL63.
- Impact on Pre-existing Immunity: The study also looked at the effect of SARS-CoV-2 vaccination on individuals with pre-existing anti-NL63 antibodies. Despite having these antibodies before vaccination, no significant increase in NL63 antibody levels was observed after mRNA-1273 or BNT162b2 vaccination. This suggests that while vaccination can generate new immune responses, it does not necessarily boost pre-existing immunity to NL63.
- Pre-existing NL63 Immunity and Vaccine Response: The study further examined whether pre-existing anti-NL63 antibodies affected the immune response to the SARS-CoV-2 vaccines. The results showed no significant correlation between the levels of pre-existing NL63 antibodies and the antibody response to SARS-CoV-2 vaccination, indicating that pre-existing immunity to NL63 does not influence the effectiveness of the SARS-CoV-2 vaccines.
- Mouse Model Studies: Mice immunized with either the Pfizer (BNT162b2) or Moderna (mRNA-1273) vaccines also showed the induction of cross-reactive antibodies. The Moderna vaccine specifically induced neutralizing antibodies against NL63, reinforcing the findings in humans.
In summary, the study indicates that SARS-CoV-2 mRNA vaccines can elicit cross-reactive antibody responses against NL63, particularly with mRNA-1273, but do not boost pre-existing immunity to NL63. This has implications for understanding the broader immune responses generated by SARS-CoV-2 vaccines and their potential effects on immunity to other seasonal coronaviruses.
Strengths of the article:
The study addresses a crucial issue in the current pandemic context, namely the need to develop vaccines that provide broader protection against various coronaviruses, including emerging ones.
Moreover, we acknowledge the reviewers’ comment regarding the potential impact of pre-existing anti-NL63 immunity on SARS-CoV-2 mRNA vaccination outcomes. Indeed, we agree that this aspect remains largely unexplored in the current literature, highlighting the need for further investigation.
Regarding the other aspects discussed, these have been addressed in previous studies. In addition to the references already cited, I would suggest including the following articles in the references:
· Amanat F, Clark J, Carreño JM, Strohmeier S, Yellin T, Meade PS, Bhavsar D, Muramatsu H, Sun W, Coughlan L, Pardi N, Krammer F. Immunity to Seasonal Coronavirus Spike Proteins Does Not Protect from SARS-CoV-2 Challenge in a Mouse Model but Has No Detrimental Effect on Protection Mediated by COVID-19 mRNA Vaccination. J Virol. 2023 Mar 30;97(3):e0166422. doi: 10.1128/jvi.01664-22.
· Singh G, Abbad A, Kleiner G, Srivastava K, Gleason C; PARIS Study Group; Carreño JM, Simon V, Krammer F. The post-COVID-19 population has a high prevalence of cross-reactive antibodies to spikes from all Orthocoronavirinae genera. mBio. 2024 Jan 16;15(1):e0225023. doi: 10.1128/mbio.02250-23.
· Amanat F, Thapa M, Lei T, Ahmed SMS, Adelsberg DC, Carreño JM, Strohmeier S, Schmitz AJ, Zafar S, Zhou JQ, Rijnink W, Alshammary H, Borcherding N, Reiche AG, Srivastava K, Sordillo EM, van Bakel H; Personalized Virology Initiative; Turner JS, Bajic G, Simon V, Ellebedy AH, Krammer F. SARS-CoV-2 mRNA vaccination induces functionally diverse antibodies to NTD, RBD, and S2. Cell. 2021 Jul 22;184(15):3936-3948.e10. doi: 10.1016/j.cell.2021.06.005.
Here are some revisions:
Lines 42-44: The statement "The potential impact of pre-existing immunity against the human coronavirus NL63 on the outcomes of mRNA vaccination for SARS-CoV-2 remains unexplored" is not entirely accurate. Previous have investigated this aspect (Amanat et al., doi: 10.1128/jvi.01664-22 and Woldemeskel et al., ref 26). Therefore, I would suggest modifying the sentence with "Although data on the effect of SARS-CoV-2 mRNA vaccines on other coronaviruses, including common cold-causing human seasonal coronaviruses like NL63, were initially limited, recent studies have begun to investigate their impact, particularly in terms of T cell cross-reactivity."
Lines 76: "Enrolment" → If you are following American English, please change it to "enrollment."
Line 78: “a history of being immunocompromised” might be more correct.
Lines 117-118: "fluorescein isothiocyanate (FITC) fluorescence versus Alexa Fluor 647 fluorescence". The word "fluorescence" might be redundant. Please change with "FITC versus Alexa Fluor 647 fluorescence."
Lines 128-129: "n = 3 replicates per dilution" à It might be more correct to write as "with three replicates per dilution (n = 3)."
Lines 137-138: “with” à “followed by”
Line 140: “Spearman’s correlation analysis”
Line 130: Seven days is a relatively long incubation time compared to most studies, which typically assess neutralization within 72 hours. It might be useful to check cell viability at an intermediate time point (e.g., 48 or 72 hours) to compare the results and determine whether the longer incubation is truly necessary. Were there specific reasons for choosing this time point? Have any differences been observed compared to the 72-hour time point?
Line 164: Please change “antibody” with “antibodies” or “antibody titer”.
Lines 166: Please replace “of” with “with”.
Line 168: “mouse” à “mice”.
Line 174, 176: regarding patients’ age, I would suggest adding 95% CI to the median age.
Line 216: “among individuals recovering”.
Author Response
C1. Strengths of the article:
The study addresses a crucial issue in the current pandemic context, namely the need to develop vaccines that provide broader protection against various coronaviruses, including emerging ones.
R: We thank the reviewer for the positive comments. We acknowledge the reviewers’ comment regarding the potential impact of pre-existing anti-NL63 immunity on SARS-CoV-2 mRNA vaccination outcomes. Indeed, we agree that this aspect remains largely unexplored in the current literature, highlighting the need for further investigation.
C2. Regarding the other aspects discussed, these have been addressed in previous studies. In addition to the references already cited, I would suggest including the following articles in the references:
Amanat F, Clark J, Carreño JM, Strohmeier S, Yellin T, Meade PS, Bhavsar D, Muramatsu H, Sun W, Coughlan L, Pardi N, Krammer F. Immunity to Seasonal Coronavirus Spike Proteins Does Not Protect from SARS-CoV-2 Challenge in a Mouse Model but Has No Detrimental Effect on Protection Mediated by COVID-19 mRNA Vaccination. J Virol. 2023 Mar 30;97(3): e0166422. doi: 10.1128/jvi.01664-22.
- Singh G, Abbad A, Kleiner G, Srivastava K, Gleason C; PARIS Study Group; Carreño JM, Simon V, Krammer F. The post-COVID-19 population has a high prevalence of cross-reactive antibodies to spikes from all Orthocoronavirinaegenera. mBio. 2024 Jan 16;15(1): e0225023. doi: 10.1128/mbio.02250-23.
- Amanat F, Thapa M, Lei T, Ahmed SMS, Adelsberg DC, Carreño JM, Strohmeier S, Schmitz AJ, Zafar S, Zhou JQ, Rijnink W, Alshammary H, Borcherding N, Reiche AG, Srivastava K, Sordillo EM, van Bakel H; Personalized Virology Initiative; Turner JS, Bajic G, Simon V, Ellebedy AH, Krammer F. SARS-CoV-2 mRNA vaccination induces functionally diverse antibodies to NTD, RBD, and S2. Cell. 2021 Jul 22;184(15):3936-3948.e10. doi: 10.1016/j.cell.2021.06.005.
R: The articles have been added to the reference list.
C3. Lines 42-44: The statement "The potential impact of pre-existing immunity against the human coronavirus NL63 on the outcomes of mRNA vaccination for SARS-CoV-2 remains unexplored" is not entirely accurate. Previous have investigated this aspect (Amanat et al., doi: 10.1128/jvi.01664-22 and Woldemeskel et al., ref 26). Therefore, I would suggest modifying the sentence with "Although data on the effect of SARS-CoV-2 mRNA vaccines on other coronaviruses, including common cold-causing human seasonal coronaviruses like NL63, were initially limited, recent studies have begun to investigate their impact, particularly in terms of T cell cross-reactivity."
R: The sentence has been modified as suggested.
C4. Lines 76: "Enrolment" → If you are following American English, please change it to "enrollment."
R: This has been amended for American English.
C5. Line 78: “a history of being immunocompromised” might be more correct.
R: This has been amended
C6. Lines 117-118: "fluorescein isothiocyanate (FITC) fluorescence versus Alexa Fluor 647 fluorescence". The word "fluorescence" might be redundant. Please change with "FITC versus Alexa Fluor 647 fluorescence."
R: This section has been corrected.
C7. Lines 128-129: "n = 3 replicates per dilution" à It might be more correct to write as "with three replicates per dilution (n = 3)."
R: This part has been amended.
C8. Lines 137-138: “with” à “followed by”
R: This part has been amended.
C9. Line 140: “Spearman’s correlation analysis”
R: This part has been amended.
C10. Line 130: Seven days is a relatively long incubation time compared to most studies, which typically assess neutralization within 72 hours. It might be useful to check cell viability at an intermediate time point (e.g., 48 or 72 hours) to compare the results and determine whether the longer incubation is truly necessary. Were there specific reasons for choosing this time point? Have any differences been observed compared to the 72-hour time point?
R: In our hands, NL63 typically requires at least 6 days to cause observable cytopathic effects (CPE) in cell culture, consistent with previous reports (Milewska et al., DOI: 10.1128/JVI.02078-14). In our experiments, 6 days of incubation produced minimal CPE, while 7 days proved optimal for observing CPE. Therefore, neutralization could not be reliably assessed at the 72-hour time point. A "no virus" control was included in each experiment to determine cell viability using the CellTiter-Glo kit.
This explanation has been added to the materials and methods section (156).
C11.Line 164: Please change “antibody” with “antibodies” or “antibody titer”.
R: This has been corrected
C12. Lines 166: Please replace “of” with “with”.
R: This has been corrected
C13. Line 168: “mouse” à “mice”.
R: This has been corrected
C14. Line 174, 176: regarding patients’ age, I would suggest adding 95% CI to the median age.
R: The median age for the PROTECT group is 33 with 95% CI 29-39; the median age for the PRIBIVAC group is 36 with 95% CI 31-59. This was added to the manuscript (line 208)
C15. Line 216: “among individuals recovering”.
R: This has been corrected
Reviewer 2 Report
Comments and Suggestions for Authors
This manuscript mainly reported how do SARS-CoV-2 mRNA vaccines and anti-NL63 seasonal coronaviruses antibodies affect each other.There are numerous major and minor concerns for improving this manuscript.
1) Similar studies have been published extensively, and the innovation of this study is not sufficient. Authors should compare their work and provide the discrepancy.
2) As known, SARS-CoV-2 belongs to the family of beta CoVs. Among four kinds of seasonal CoVs ,OC43 and HKU1 belong to beta CoVs,but NL63 and 229E belong to alpha CoVs. So, from a genetic evolution, SARS-CoV-2 and (OC43 or HKU1)have a closer genetic relationship than NL63, and thus it is not strange that SARS-CoV-2 vaccination did not boost pre-existing anti-NL63 responses. Authors should detect whether SARS-CoV-2 vaccination can boost pre-existing anti-OC43 or HKU1 responses. Authors should reference the following similar studies, such as PMID: 37654488; PMID: 36159791; PMID: 36539527; et al.
3) Line 68: Figures should be numbered in the order they appear, and cannot start with Figure 2.
4) Figure 1: why mRNA-1273 vaccination induced a significantly higher level of anti-NL63 spike antibodies, but BNT162b2 can not? Since both of them are SARS-CoV-2 Spike-based mRNA vaccine. Should be listed the meaning of **** in the figure legend.
5) LIne 175: “vilunteers”should be“volunteers”
6) Line 180: it is strange for “anti-NL63 levels decreased after the third BNT162b2 dose”.
7) Figure 2: Why use BNT162b2 vaccinated samples for this study, since BNT162b2 can not induce a significantly high level of anti-NL63 spike antibodies as shown in the Figure 1? Why not use mRNA-1273 vaccinated samples?
Comments on the Quality of English LanguageThe English could be improved to more clearly express the research.
Author Response
1) Similar studies have been published extensively, and the innovation of this study is not sufficient. Authors should compare their work and provide the discrepancy.
While studies have examined the effect of SARS-CoV-2 vaccination on other coronaviruses, the effect of NL63 on SARS-CoV-2 vaccine immunogenicity has not been extensively studied. Furthermore, the influence of pre-existing anti-NL63 antibodies on SARS-CoV-2 vaccine immunogenicity has not been studied in mice and humans. This is exemplified by the data provided in the Table.
2) As known, SARS-CoV-2 belongs to the family of beta CoVs. Among four kinds of seasonal CoVs, OC43 and HKU1 belong to beta CoVs, but NL63 and 229E belong to alpha CoVs. So, from a genetic evolution, SARS-CoV-2 and (OC43 or HKU1) have a closer genetic relationship than NL63, and thus it is not strange that SARS-CoV-2 vaccination did not boost pre-existing anti-NL63 responses. Authors should detect whether SARS-CoV-2 vaccination can boost pre-existing anti-OC43 or HKU1 responses. Authors should reference the following similar studies, such as PMID: 37654488; PMID: 36159791; PMID: 36539527; et al.
The articles mentioned above have been added to the manuscript. However, we want to emphasize that although SARS-CoV-2 and NL63 are distinct viruses, their spike proteins both bind to the ACE2 receptor. We hypothesized that an immune response against one spike protein might influence the immune response against the other.
3) Line 68: Figures should be numbered in the order they appear and cannot start with Figure 2.
The figure numbering has been adjusted.
4) Figure 1: why mRNA-1273 vaccination induced a significantly higher level of anti-NL63 spike antibodies, but BNT162b2 can not? Since both of them are SARS-CoV-2 Spike-based mRNA vaccine. Should be listed the meaning of **** in the figure legend.
While these two vaccines share the same amino acid sequence for the spike protein, they differ in their lipid nanoparticle formulations and mRNA modifications. These differences may affect the immunogenicity and the breadth of the antibody response, as observed previously when assessing breadth against SARS-CoV-2 variants (doi:10.1038/s41586-021-04186-8). This was discussed in the discussion section (line 278).
The meaning of “****” has been added in the figure legend.
5) Line 175: “vilunteers” should be “volunteers”
The typo has been corrected.
6) Line 180: it is strange for “anti-NL63 levels decreased after the third BNT162b2 dose”.
We agree with the reviewer that the decrease in NL63 antibody levels after the third dose is somewhat surprising. However, we believe this observation demonstrates a waning of the anti-NL63 antibody response and further confirms that the BNT162b2 vaccine has no significant influence on anti-NL63 responses. This point has been added to the revised manuscript (line 215).
7) Figure 2: Why use BNT162b2 vaccinated samples for this study, since BNT162b2 cannot induce a significantly high level of anti-NL63 spike antibodies as shown in the Figure 1? Why not use mRNA-1273 vaccinated samples?
In Figure 1, the mouse study focused on whether vaccination can induce a cross-reactive antibody response against NL63. However, in Figure 2, the human study focused on whether vaccination can boost pre-existing anti-NL63 responses, as NL63 circulates in the population, and most of our volunteers had detectable pre-existing anti-NL63 antibodies (Figure 3). We were unable to obtain samples from individuals vaccinated only with mRNA-1273. Therefore, we recruited a cohort of individuals who received an mRNA-1273 booster and compared samples before and after booster administration. We added a few lines to the discussion section as a limit of our study (line 295).
- The English could be improved to more clearly express the research.
The text has been edited for typos and grammar.
Round 2
Reviewer 1 Report
Comments and Suggestions for Authors
Dear Authors,
I would like to confirm that the revisions I requested have been properly addressed. The changes appear to have been made accurately.
Author Response
Thanks!